# Targeted genome editing *in vivo* corrects a *Dmd* duplication restoring wild-type dystrophin expression

Eleonora Maino[1,2,†] [iD], Daria Wojtal[1,2,†], Sonia L Evagelou[1], Aiman Farheen[1], Tatianna W Y Wong[1,2], Kyle Lindsay[1], Ori Scott[1,3,4], Samar Z Rizvi[1,2], Elzbieta Hyatt[1], Matthew Rok[1,2], Shagana Visuvanathan[1], Amanda Chiodo[1], Michelle Schneeweiss[1], Evgueni A Ivakine[1,5,‡] & Ronald D Cohn[1,2,3,4,*,‡] [iD]

## Abstract

Tandem duplication mutations are increasingly found to be the direct cause of many rare heritable diseases, accounting for up to 10% of cases. Unfortunately, animal models recapitulating such mutations are scarce, limiting our ability to study them and develop genome editing therapies. Here, we describe the generation of a novel duplication mouse model, harboring a multi-exonic tandem duplication in the *Dmd* gene which recapitulates a human mutation. Duplication correction of this mouse was achieved by implementing a single-guide RNA (sgRNA) CRISPR/Cas9 approach. This strategy precisely removed a duplication mutation *in vivo*, restored full-length dystrophin expression, and was accompanied by improvements in both histopathological and clinical phenotypes. We conclude that CRISPR/Cas9 represents a powerful tool to accurately model and treat tandem duplication mutations. Our findings will open new avenues of research for exploring the study and therapeutics of duplication disorders.

**Keywords** AAVs; CRISPR/Cas9; Duchenne muscular dystrophy; duplication mutations; genome editing

**Subject Categories** Genetics, Gene Therapy & Genetic Disease; Musculoskeletal System

## Introduction

Complex structural rearrangements are increasingly recognized as causing human genetic disease. In particular, duplication mutations are estimated to account for almost 10% of all rare inherited disorders (Rees & Liu, 2018). Pathogenic duplication mutations, most notably tandem duplications, may occur at multiple levels, ranging from intragenic duplications (duplications of a region within a single gene), to full single-gene or multi-gene duplications. Accordingly, the spectrum of duplication-related disorders is broad. Duplication of chromosomal regions involving multiple genes underlie MeCP2 duplication syndrome (MDS) (Van Esch, 2011), Pelizaeus-Merzbacher Disease (PMD; Osorio & Goldman, 2018), and Charcot-Marie Tooth (CMT; Lupski, 1999). Intragenic duplications, which typically result in disruption of the open reading frame (ORF) and loss-of-function of the affected protein, have been implicated in a number of currently incurable diseases such as Duchenne muscular dystrophy (DMD; Bladen *et al*, 2015), ataxia telangiectasia (Cavalieri *et al*, 2007), and Alport syndrome (Arrondel *et al*, 2004).

The study and treatment of duplication mutations presents unique challenges, and model organisms recapitulating such mutations are limited. Duplication disorders, such as MDS, PMD, and CMT, have been primarily investigated using transgenic animals overexpressing the causative genes (Kagawa *et al*, 1994; Magyar *et al*, 1996; Collins *et al*, 2004). In addition, chromosomal engineering via the Cre-Lox system and, more recently, genome engineering have been used to generate some mouse models involving tandem multi-gene duplications (Walz *et al*, 2006; Li *et al*, 2007; Nakatani *et al*, 2009; Yu *et al*, 2010; Horev *et al*, 2011; Clark *et al*, 2013; Pristyazhnyuk *et al*, 2019). With regard to intragenic duplications, the only available model to date is the *Dup2* model affecting the *Dmd* gene (Vulin *et al*, 2015). The *Dup2* mouse model recapitulates a duplication of the *Dmd* exon 2 at the transcript level and has been crucial for the development of novel DMD therapeutic approaches (Wein *et al*, 2014). However, the *Dup2* mouse model does not fully recapitulate the sequence topology typical of tandem duplications at

1 Program in Genetics and Genome Biology, the Hospital for Sick Children Research Institute, Toronto, ON, Canada
2 Department of Molecular Genetics, University of Toronto, Toronto, ON, Canada
3 Institute of Medical Science, University of Toronto, Toronto, ON, Canada
4 Department of Pediatrics, the Hospital for Sick Children, Toronto, ON, Canada
5 Department of Physiology, University of Toronto, Toronto, ON, Canada
*Corresponding author. Tel: +1 416 8137654 (327115); E-mail: ronald.cohn@sickkids.ca
†These authors contributed equally to this work
‡These authors contributed equally to this work as senior authors

the genomic level since it was generated by knocking-in a cassette containing exon 2 and its immediate intronic region in the *Dmd* intron 2. In this regard, the scarcity of animal models faithfully modeling tandem intragenic duplications has limited the testing and development of genome-editing therapies targeting such mutations.

In an effort to study novel genome-editing therapies broadly applicable to tandem duplications, we focused on DMD as a disease model. DMD is caused by the lack of dystrophin expression due to mutations disrupting the *DMD* ORF (Hoffman *et al*, 1987). Thousands of different mutations have been described causing the disease, including large deletions and duplications, point mutations, and small rearrangements. Up to 10–15% of DMD patients harbor single or multi-exon *DMD* duplications (Bladen *et al*, 2015). Recent *in vitro* work (Wojtal *et al*, 2016; Lattanzi *et al*, 2017; Long *et al*, 2018) by our group and others has shown that a single-sgRNA CRISPR/Cas9 strategy can be utilized to correct duplication mutations and restore dystrophin expression in patient cells with DMD duplications. However, *in vivo* testing of our strategy was previously hampered by the lack of appropriate animal models.

Herein, we describe the generation of the first intragenic tandem duplication mouse model recapitulating a patient mutation using CRISPR/Cas9. The *Dup18-30* mouse harbors a 137 kb multi-exonic duplication of the exons from 18 to 30 in the *Dmd* gene and shows a robust DMD phenotype. In addition, we present the *in vivo* correction of a duplication mutation utilizing the single-sgRNA approach, resulting in full-length dystrophin restoration, and significant phenotypic improvement. Our work opens new possibilities in modeling and correcting tandem duplication mutations, which may be applied to a wide array of duplication-related disorders.

# Results

### A two-step CRISPR/Cas9 strategy generates the first intragenic tandem multi-exonic duplication mouse model

To generate an intragenic duplication mouse model having a duplication of the exons from 18 to 30 in the *Dmd* gene, we designed four sgRNAs referred to as i17A, i17B, i30A, and i30B, targeting introns 17 and 30 flanking the region to duplicate (Fig 1A). The guides were introduced into mouse embryonic stem cells (mESC) by means of electroporation, followed by clone screening via PCR to identify the duplication junction (Appendix Fig S1). Of the 243 screened clones, three positive clones (1.23%) were expanded, aggregated, and injected into blastocysts which were implanted in pseudo-pregnant mice. Of the three mice positive for the duplication junction, one showed no evidence of germline transmission of the duplication, and another demonstrated a complex rearrangement in lieu of a duplication. Accordingly, these two mice were excluded from further analysis.

As CRISPR/Cas9 editing may generate inadvertent structural variants, the founder mouse was analyzed via whole genome sequencing (WGS), confirming the presence of the predicted 136.8 kb duplication (ChrX: 83,737,872–83,874,709). However, the second copy of the duplication was immediately followed by an unwanted 12,049 bp inversion of the region spanning introns 30 to 34 (ChrX: 83,876,785–83,888,834) (Appendix Table S1). This mouse model, referred as *Dup18-30i*, presented *Dmd* splicing abnormalities

(Appendix Fig S2A and B), lack of dystrophin expression, and compromised muscle physiology (Appendix Fig S3A and B).

To correct the inversion, we designed two gRNAs flanking the inverted DNA region (Fig 1B), which were electroporated together with the Cas9 protein into *Dup18-30i* zygotes. We screened newborn mice and detected the predicted re-inversion junctions in 6.25% of them. One founder, the *Dup18-30* mouse model, exhibited germline transmission of the re-inverted allele together with the duplication junction. WGS confirmed that the inversion was corrected without any alteration of the tandem duplication (Appendix Table S2).

Molecular analysis via PCR and Sanger sequencing of the duplication junction in the *Dup18-30* mouse model revealed joining of intron 30 and 17, along with a 96-bp intronic deletion (Fig 1C). Additionally, RT–PCR analysis of the *Dup18-30 Dmd* transcript showed the presence of the predicted 2065 bp duplication of the exons from 18 to 30 at the RNA level (Fig 1D, Appendix Fig S2A and C). The correct joining of exons 30 and 18 was confirmed by Sanger sequencing the *Dup18-30* cDNA (Fig 1D). Western blotting analysis revealed absent dystrophin expression in skeletal and cardiac muscles of *Dup18-30* mice (Fig 1E). These findings confirmed *Dup18-30* to be the first *Dmd* multi-exonic tandem duplication mouse model faithfully recapitulating a patient mutation.

### The *Dup18-30* mouse model recapitulates DMD disease manifestations

Immunohistochemical analysis of cardiac and skeletal muscles in 15-week-old *Dup18-30* mice showed complete absence of dystrophin expression except for a few revertant fibers (RFs) (Figs 2A and EV1A). *Dup18-30* muscles showed sporadic clusters of RFs, dystrophin-positive fibers that arise from spontaneous exon skipping events, commonly reported in DMD patients and animal models (Pigozzo *et al*, 2013). The tibialis anterior (TA) and triceps showed spontaneous dystrophin expression in 4.7% and 2.3% of the fibers, respectively (Fig 2C).

The lack of dystrophin expression resulted in dystrophic muscle architecture, fibrosis, central nuclei, and heterogeneous fiber size which are typical signs of muscular dystrophy (Figs 2B and EV1B). TA and triceps samples of *Dup18-30* mice demonstrated 65.4 and 75.9% central nuclei (compared to 0% in WT), respectively (Fig 2D). The muscle strength of the *Dup18-30* mice was evaluated with forelimbs grip strength and specific tetanic force measurements in the TA muscle. The grip strength and the tetanic force were decreased by 22.2% ($P < 0.001$) and 40.5% ($P < 0.001$) in *Dup18-30* mice compared to WT, respectively (Fig 2E–F). The locomotor function of the *Dup18-30* was assessed via open-field test. The *Dup18-30* mice exhibited decreased performance compared to their WT counterparts with respect to total distance traveled ($P < 0.001$), vertical activity ($P < 0.001$), average speed ($P < 0.001$), and total resting time ($P < 0.01$) (Figs 2G and H, and EV1C and D).

### A single-sgRNA/Cas9 treatment removes the *Dmd* duplication and restores full-length dystrophin expression in *Dup18-30* mice

We employed the single-sgRNA approach, previously used successfully *in vitro* (Wojtal *et al*, 2016) to remove the duplication in the *Dup18-30* mice *in vivo*. The approach consists of a single-sgRNA guide that together with a Cas9 targets both copies of the *Dmd*

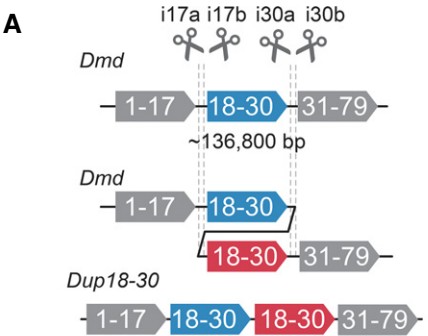

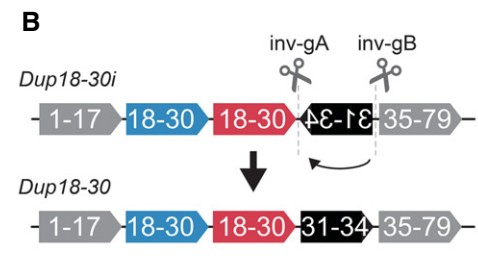

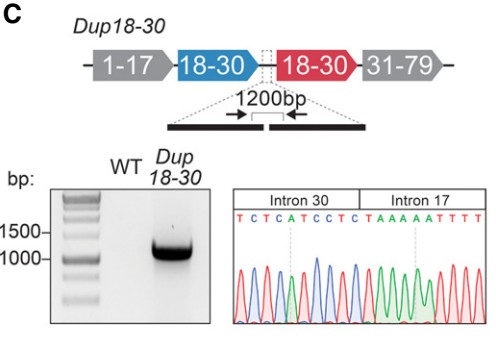

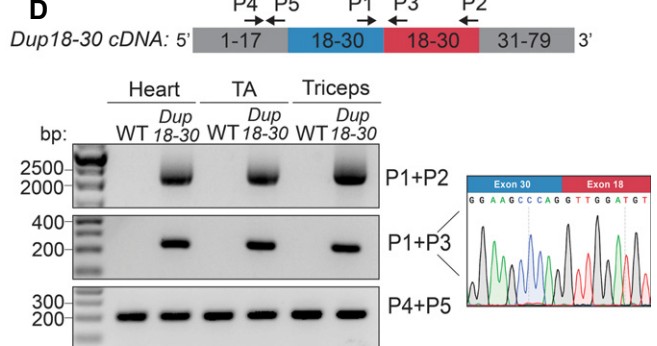

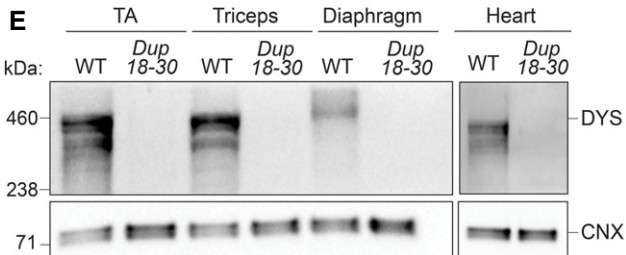

**Figure 1. A CRISPR/Cas9-based strategy generates a mouse model exhibiting a 137 kb multi-exonic tandem head-to-tail duplication in the *Dmd* gene.**

A   A 137 kb region encompassing exons 18–30 was targeted with 4 sgRNAs in introns 17 and 30 to generate a genomic duplication. Cas9 and the sgRNAs are represented as scissors.

B   Schematics of the 12,049 bp inversion present in the *Dup18-30i* mice and of the strategy utilized to correct the inversion. Two sgRNAs targeting the inversion were utilized to rescue the complex rearrangement. Cas9 and the sgRNAs are represented as scissors.

C   PCR amplification and Sanger sequencing of the duplication junction in the *Dup18-30* founder mouse confirming the joining of intron 30 and 17, along with a 96 bp intronic deletion.

D   The heart, tibialis anterior (TA), and triceps muscle were isolated and analyzed to identify the presence of the duplication from exons 18 to 30 via RT–PCR using primers represented by the arrows. Sanger sequencing confirmed correct splicing of exons 30 and 18 at the duplication junction.

E   Protein lysates isolated from the TA, triceps, diaphragm, and heart muscles of WT and *Dup18-30* mice were probed for dystrophin expression by Western blot. Calnexin serves as a loading control.

duplication. Upon Cas9 cleavage and re-ligation of the DNA ends, the region encompassed between the two single-sgRNA target sites is removed, restoring the *Dmd* ORF and dystrophin expression (Fig 3A). To select the single-sgRNA to treat the *Dup18-30* mice, we scanned all introns within the duplicated region. We then selected the best sgRNAs based on their off-target scores, while verifying that

they would not interfere with any predicted splice sites. The top eight-ranking guides were tested *in vitro* in N2A cells, and the most active guide, a sgRNA targeting intron 21 of the *Dmd* gene (i21), was chosen for the *in vivo* treatment (Appendix Table S3). A *Staphylococcus aureus* Cas9 (SaCas9) driven by the constitutive CMV promoter and the i21 sgRNA encoding cassettes were

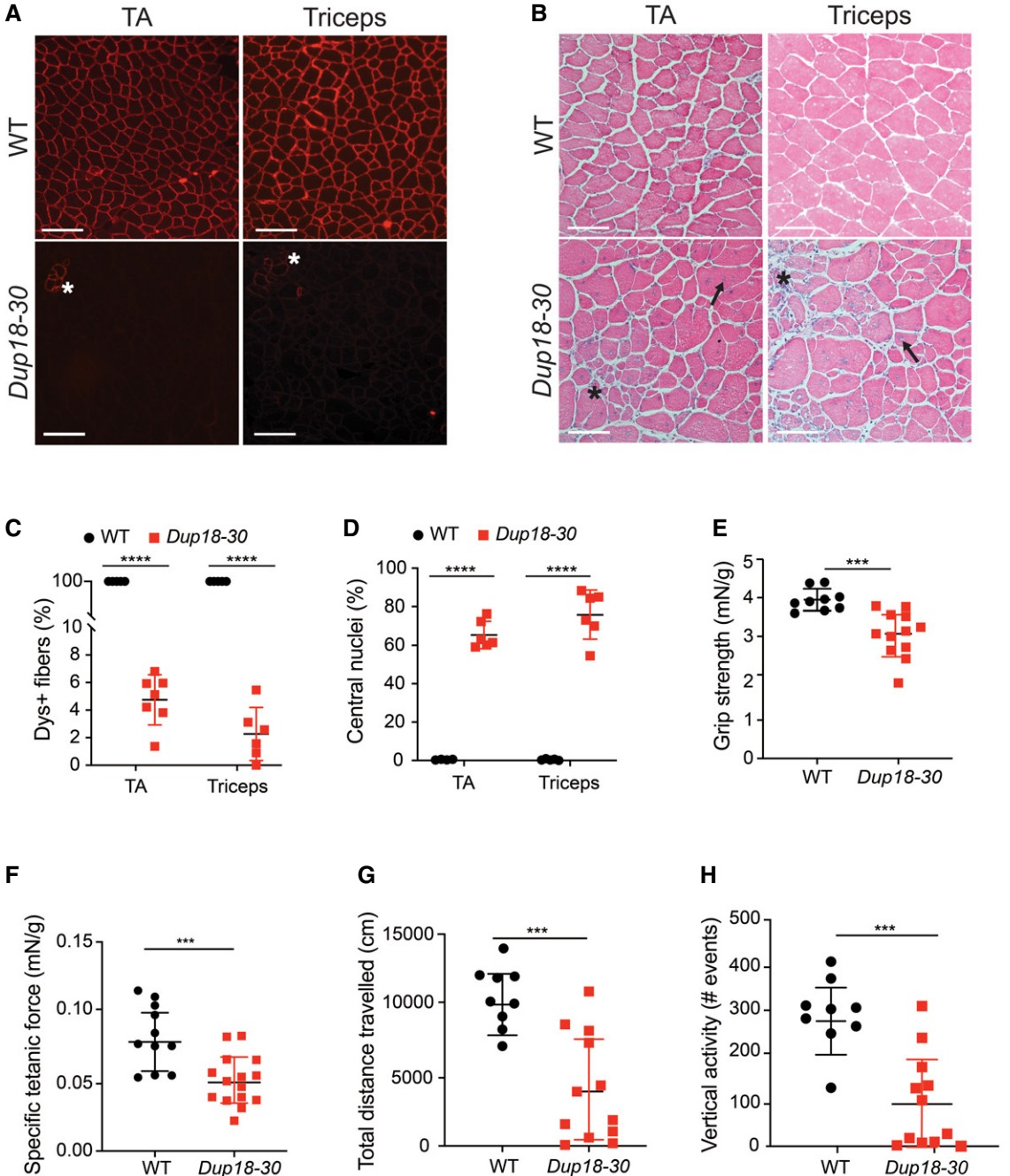

**Figure 2. The *Dup18-30* mouse model recapitulates DMD disease manifestations.**

A  8-μm cross section of 15-week-old WT and *Dup18-30* TA and triceps were analyzed for dystrophin localization by immunofluorescence. Asterisk indicates clusters of revertant fibers. Scale bars, 100 μm.

B  The muscle architecture of the same muscles was investigated by H&E staining. A representative image is shown. Scale bars, 100 μm. Asterisk indicates areas with necrotic fibers and fibrosis. Arrows indicate fibers with central nuclei.

C  Percentage of dystrophin-positive fibers in the TA and triceps of WT and *Dup18-30* mice. WT, *n* = 4; *Dup18-30*, *n* = 6–7.

D  Percentage of myofibers with centrally located nuclei in the TA and triceps of WT and *Dup18-30* mice. WT, *n* = 4; *Dup18-30*, *n* = 6.

E  Forelimb grip strength was measured in 15-week-old WT and *Dup18-30* mice. WT, *n* = 9; *Dup18-30*, *n* = 12.

F  Specific tetanic force was measured in 15-week-old mice using an *in vivo* muscle-function analyzer. WT, *n* = 11; *Dup18-30*, *n* = 15 mice.

G, H  Mice were tested in an open-field chamber in which total distance traveled (G) and vertical activity (H) were assessed. WT, *n* = 9; *Dup18-30*, *n* = 12.

Data information: Data are represented as means ± SD. Statistical analyses were performed with Student's *t*-test. ****P* < 0.001, *****P* < 0.0001.

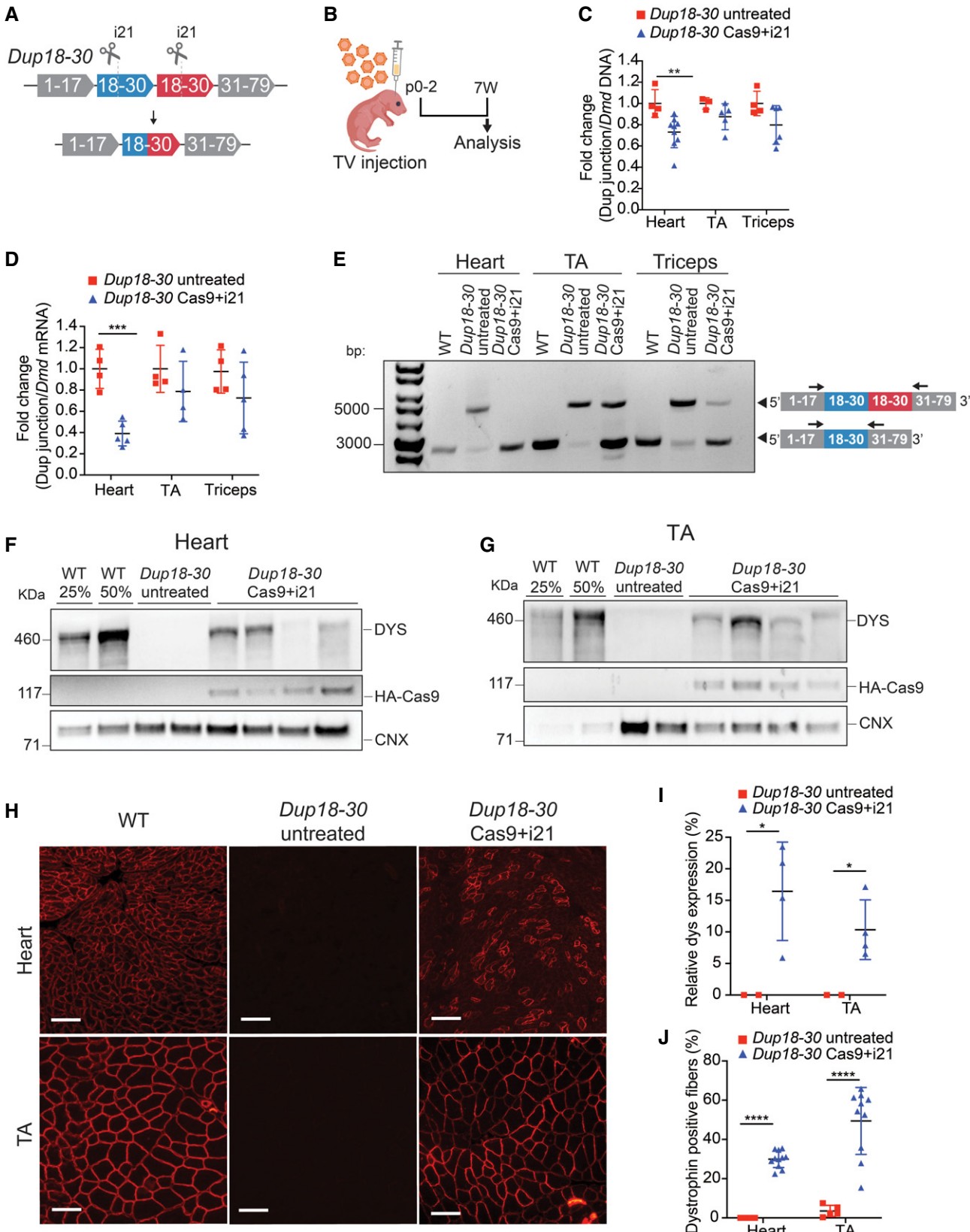

Figure 3.

**Figure 3. A single-sgRNA/Cas9 treatment removes the *Dmd* duplication and restores full-length dystrophin expression in the *Dup18-30* mouse model.**

A  Schematic of the single-sgRNA CRISPR/Cas9 strategy to remove the duplication of the exons from 18 to 30 in the *Dup18-30* mouse model. A single-sgRNA targeting intron 21 (i21) was utilized to cut both copies of the duplicated region. Cas9 and the sgRNAs are represented as scissors.

B  Two-day-old neonatal *Dup18-30* pups were injected with AAV9 carrying Cas9 and i21 gRNA (n = 11; $3 \times 10^{12}$ viral genomes) via temporal vein and sacrificed 7 weeks later.

C  The efficiency of the removal of the duplicated region at the DNA level was assessed via qPCR analysis by normalizing the signal obtained from the duplication junction to that of the total *Dmd* signal. *Dup18-30* untreated, n = 3–4; *Dup18-30* treated, n = 5–9.

D  The RNA editing efficiency was quantified in the same tissues analyzed in panel B via qPCR utilizing the expression ratio between the duplication junction and the WT *Dmd* transcript. *Dup18-30* untreated, n = 4; *Dup18-30* treated, n = 4–5.

E  RT–PCR analyzing the removal of the duplication from the heart, TA and triceps muscle of WT, *Dup18-30*-untreated, and *Dup18-30*-treated mice. Arrows correspond to primers in exon 17 and exon 33.

F, G  Western blotting detected restoration of dystrophin expression in (F) heart and (G) TA in the *Dup18-30* mice. 25% and 50% of the WT proteins compared to *Dup18-30* mice have been loaded on the gel. Calnexin was used as a loading control.

H  Immunostaining showed restoration of dystrophin expression in the heart and TA. A representative sample is shown. Scale bars, 100 µm.

I  Quantification of dystrophin Western blot in panels F and G. *Dup18-30* untreated, n = 2; *Dup18-30* treated, n = 4.

J  Percentage of dystrophin-positive fibers from the immunostaining in the heart and TA of *Dup18-30*-untreated and *Dup18-30*-treated mice. *Dup18-30* untreated, n = 5; *Dup18-30* treated, n = 10–11.

Data information: All data are represented as the mean ± SD. Statistical analyses were performed with Student's *t*-test. *$P < 0.05$, **$P < 0.01$, ***$P < 0.001$, ****$P < 0.0001$.

packaged into adeno-associated virus 9 (AAV9), which has a known tropism for skeletal and cardiac muscle (Lau & Suh, 2017). The AAV9-Cas9-i21sgRNA viral particles were delivered to neonatal mice in the first 2 days of life via temporal vein injection at a dosage of $3 \times 10^{12}$ VG (viral genomes) per mouse (Fig 3B). Outcomes were analyzed 7 weeks post-injection.

We analyzed the efficacy of duplication removal at the DNA level by qPCR, comparing the region encompassing the duplication junction, and a normal intron-exon junction (outside the duplicated region). The treatment group showed an average 26.92% reduction in the presence of the duplication junction in the heart, translating into editing of 9.5–58.4% ($P < 0.01$) in individual mice (Fig 3C). A similar trend was noted for TA (12.22%; $P = 0.15$) and triceps (20.2%; $P = 0.085$), with editing efficiencies of up to 30% and 42% in individual mice, respectively (Fig 3C). Additionally, targeted deep amplicon sequencing of the sgRNA target site in the heart and TA muscles revealed on average 5.7% and 3.1% indels formation, respectively (Appendix Fig S4A–D). Deep sequencing of the top 11 potential off-target sites did not show any non-specific activity of the i21 sgRNA in *Dup18-30* Cas9 + i21 treated compared to untreated mice (Appendix Table S4).

The efficiency of editing at the RNA level was further assessed via qRT–PCR. A 60.9% reduction in the presence of the duplication was noted in the hearts of treated compared to untreated mice ($P < 0.001$) (Fig 3D). A similar trend was noted for TA (19.39%; $P = 0.28$) and triceps (27.3%; $P = 0.24$) (Fig 3D). Editing efficiency in individual mice was found to be up to 46% in TA and up to 62% in triceps at the RNA level. Furthermore, RT–PCR analysis of the *Dmd* transcript showed removal of the duplicated region and restoration of the wild-type *Dmd* transcript in heart and skeletal muscles (Fig 3E).

Seven weeks following treatment, dystrophin level in hearts of treated mice, as measured by Western blotting, was found to be on average 16.42% of WT [range: 5–25%; 0% in untreated mice ($P < 0.05$)] (Fig 3F and I). Dystrophin expression improved in other tissues as well, ranging from 4 to 18% of WT dystrophin protein in the diaphragm, TA, and triceps of treated mice ($P < 0.05$ compared to untreated mice) (Figs 3G and I, and EV2A–C). The percentage of dystrophin-positive fibers was further assessed by immunofluorescence, ranging from 22 to 36% for heart, and 12–68% for skeletal muscle ($P < 0.0001$ compared to untreated mice) (Figs 3H and J, and EV2D and E).

## CRISPR/Cas9-mediated duplication correction improves dystrophic phenotypes in the *Dup18-30* mice

To further investigate treatment efficacy, we performed immunostaining for components of the dystrophin-associated glycoprotein complex (DGC). In DMD, lack of dystrophin destabilizes the DGC, resulting in loss of normal muscle architecture (Campbell & Kahl, 1989; Ervasti *et al*, 1990). As expected, untreated *Dup18-30* mice revealed absent localization of DGC components at the sarcolemma. Treated mice showed partially restored expression of the DGC components, including alpha-syntrophin, beta-sarcoglycan, and neuronal nitric oxide synthase (nNOS) across all muscle samples analyzed (Figs 4A and EV3A and B). H&E staining performed on TA, triceps, and diaphragm of *Dup18-30*-treated mice showed overall improved muscle pathology with a reduction in the typical hallmarks of dystrophic muscles, including infiltration, fibrosis, and central nuclei (Fig 4B). In regard to central nuclei, a 68.5% reduction compared to the untreated group was noted in the TA and diaphragm ($P < 0.0001$), with a 60.3% reduction noted in triceps ($P < 0.001$) (Fig 4C). Further analysis of the treatment showed a beneficial bystander effect exerted by dystrophin restoration on dystrophin-negative unedited fibers. In treated mice, only 14.7% of dystrophin-negative fibers presented central nuclei, marking a 75.4% reduction in central nuclei in dystrophin-negative fibers compared with untreated mice (Fig EV4A and B).

We subsequently assessed the diaphragm, which is severely affected in DMD patients representing a substantial cause of morbidity and mortality. Masson's Trichrome staining was performed to analyze diaphragmatic fibrosis, showing a 61.2% reduction of fibrosis in *Dup18-30*-treated mice compared with untreated controls (Fig 5A and B). In addition, we evaluated forelimb grip strength and contractile force measurements in the TA. In the treated group, grip strength was increased by 75.3% ($P < 0.01$) (Fig 5C), while specific tetanic force was 48.4% higher compared to untreated mice ($P < 0.01$) (Fig 5D). Notably, specific tetanic force was not

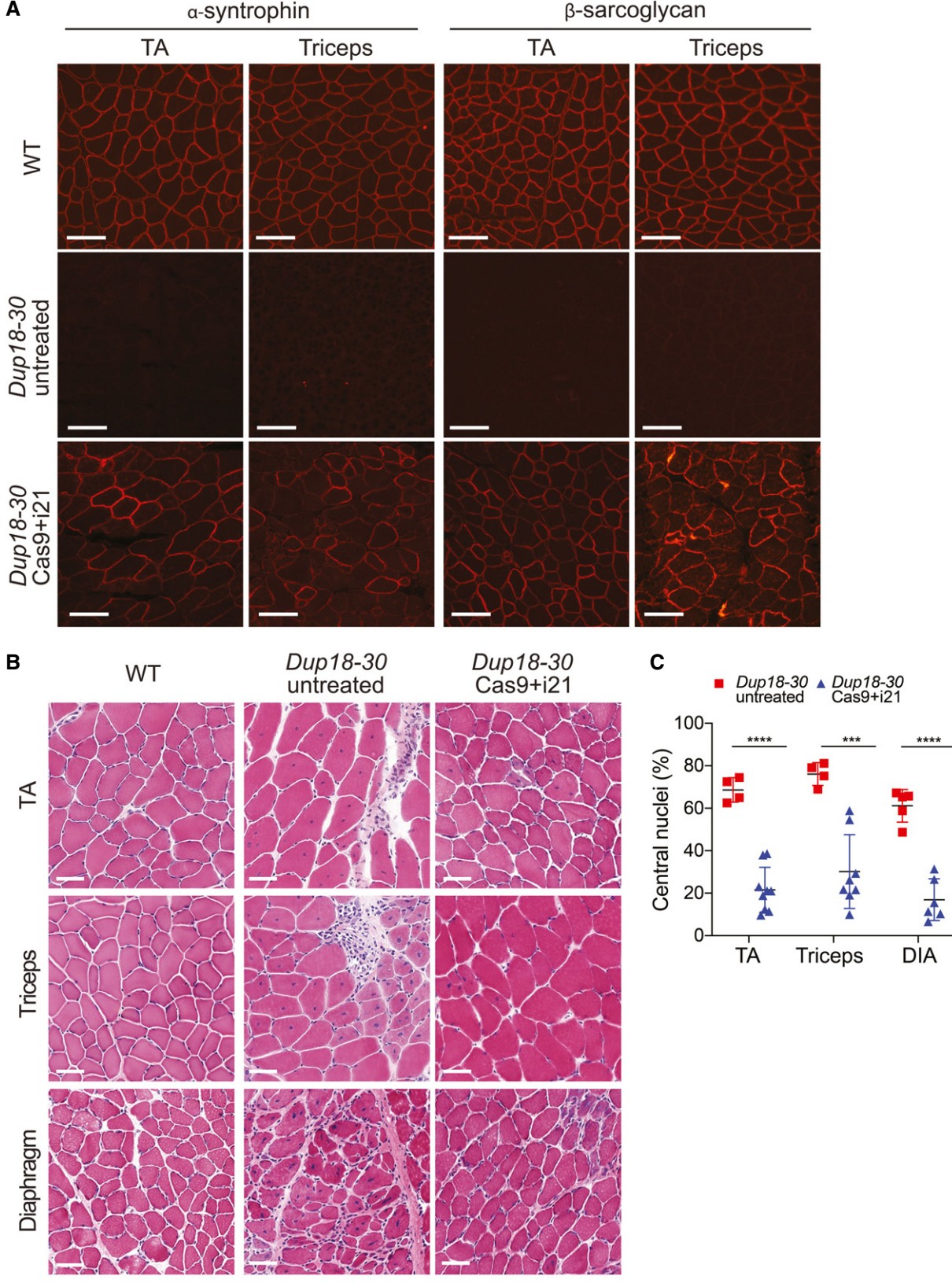

**Figure 4.**

**Figure 4. The single-sgRNA/Cas9 treatment restores DGC expression and improves muscle pathology in *Dup18-30* mice.**

A  Immunofluorescence staining for alpha-syntrophin and beta-sarcoglycan in the TA and triceps muscles of the *Dup18-30* mice. A representative image is shown. Scale bars, 100 μm.
B  TA, triceps, and diaphragm muscle architecture were analyzed by H&E staining. Scale bars, 50 μm.
C  Central nuclei were quantified in the TA, triceps, and diaphragm muscles. *Dup18-30* untreated, $n = 4–5$; *Dup18-30* treated, $n = 6–9$. All data are represented as means ± SD. Statistical analyses were performed with Student's *t*-test. ***$P < 0.001$, ****$P < 0.0001$.

significantly different between the WT and treated group ($P = 0.98$). We evaluated our treated mice functionally via open-field testing, and they demonstrated WT-like performance across all parameters including total distance traveled ($P = 0.11$), vertical activity ($P = 0.072$), total resting time ($P = 0.64$), and average speed ($P = 0.30$). This represents a significant improvement in all open-field parameters compared to untreated mice [total distance traveled ($P < 0.0001$), vertical activity ($P < 0.0001$), total resting time ($P < 0.001$), and average speed ($P < 0.05$)] (Figs 5E and F, and EV5A and B).

## Discussion

In recent years, mouse models of DMD have proven instrumental in developing and studying genome-based therapies for DMD. To date, the most commonly utilized DMD mouse model has been the *mdx* mouse model, which carries a spontaneous nonsense point mutation in exon 23 leading to absent dystrophin expression (Sicinski *et al*, 1989). In recent years, other DMD mouse models recapitulating patient-specific deletions have been generated. These models carry deletions of *DMD* exons, including single (Amoasii *et al*, 2017; Young *et al*, 2017; Min *et al*, 2019a; Min *et al*, 2020) and multi-exonic deletions (Egorova *et al*, 2019; Wong *et al*, 2020). Concerning duplication mutations, the only model previously available was *Dup2* (Vulin *et al*, 2015) harboring a single exon duplication of *Dmd* exon 2.

In the current study, we successfully generated the first-of-its-kind multi-exonic intragenic tandem duplication mouse model, recapitulating a known DMD patient mutation spanning exons 18 to 30. The *Dup18-30* mouse displays a dystrophic phenotype, comparable to that of *mdx* mice with respect to various histopathological hallmarks of muscle dystrophy, muscle strength, and locomotor parameters (Duddy *et al*, 2015; van Putten *et al*, 2020). At 15 weeks of age, the *Dup18-30* mouse model showed that 65–80% of myofibers had central nuclei, indicating the presence of cycles of necrosis and regeneration that are typically reported in dystrophin-deficient mouse models in early disease stages. The diaphragm of the *Dup18-30* mice presented elevated levels of fibrosis as early as 7 weeks of age demonstrating critical involvement of the diaphragm in our model, as seen in patients (McGreevy *et al*, 2015). One of the hallmarks of DMD in humans is the development of cardiomyopathy, with progression to heart failure being a leading cause of mortality (Fayssoil *et al*, 2010). This feature is not well-recapitulated in *mdx* mice, which generally start showing a mild cardiac phenotype in late stage of disease (Chu *et al*, 2002; Quinlan *et al*, 2004; Au *et al*, 2011). More recently, the *Dmd ΔEx52-54* mouse model was the first to show an early onset dystrophic cardiac phenotype and cardiac functional abnormalities that closely recapitulate the human disease (Wong *et al*, 2020). Further analysis in the *Dup18-30* mouse model

would be required to assess cardiac function and investigate progression of disease manifestations in late disease stages.

Our mouse model generation process via a modified CRISVAR protocol (Kraft *et al*, 2015) reflects some of the challenges involved in modeling tandem duplications. Our initial choice of employing multiple sgRNAs to cleave the duplicated region has been proposed to enhance the likelihood of obtaining the desired rearrangement (Boroviak *et al*, 2016). Indeed, we observed the presence of duplication junctions in approximately 1% of the screened clones, confirming the intrinsic complexity of generating this structural variant (Kraft *et al*, 2015; Boroviak *et al*, 2016; Pristyazhnyuk *et al*, 2019). However, achieving the desired duplication was accompanied by a second, undesired structural variation, as previously described by others (Shin *et al*, 2017; Kosicki *et al*, 2018). This underscores the importance of thoroughly analyzing the genome of newly generated mouse models, as unwanted complex rearrangements could be missed when genotyping using PCR-based assays.

Therapeutic options for heritable disorders caused by duplication mutations are scarce, and mostly limited to supportive measures and tertiary prevention. In regards to whole-gene duplication disorders such as MDS, PMD, and CMT, current experimental therapies involve reducing gene expression using anti-sense oligonucleotides (ASOs) or micro-RNAs (Sztainberg *et al*, 2015; Zhao *et al*, 2018; Li *et al*, 2019). However, these approaches do not exclude the possibility of over-targeting and thereby detrimentally lowering the expression of the duplicated genes to a level below the physiological threshold. With respect to intragenic duplication disorders, the use of ASO has been proposed to induce exon skipping of only one copy of the duplicated exons in the context of DMD (Aartsma-Rus *et al*, 2004; Aartsma-Rus *et al*, 2007; Forrest *et al*, 2010; Greer *et al*, 2014; Wein *et al*, 2017). These ASO-based approaches carry a number of important limitations. Primarily, continuous administration would be required to sustain the benefits of the treatment. Indeed, clinical trial data from DMD deletion patients treated with ASOs show very low levels of dystrophin expression (< 1%) after one year of treatment (Charleston *et al*, 2018; Frank *et al*, 2020). As for exon skipping, such treatment may result in the skipping of both copies of the duplicated region, resulting in an out-of-frame deletion. Furthermore, approximately 70% of *DMD* duplications involve multiple exons, necessitating the use of multiple ASOs for exon skipping (Flanigan *et al*, 2009). Importantly, while the above proposed therapies are aimed at ameliorating symptoms and prolonging longevity, they are not curative in principle and do not result in restoration of full-length dystrophin expression.

The single-sgRNA CRISPR/Cas9 approach has several benefits: a small number of treatment components packaged into one AAV, precise mutation targeting, and significant phenotypic improvement due to restoration of full-length dystrophin. Studies in dystrophin-negative female mice with skewed chromosome X-inactivation suggest that 3–14% of full-length dystrophin can improve muscle

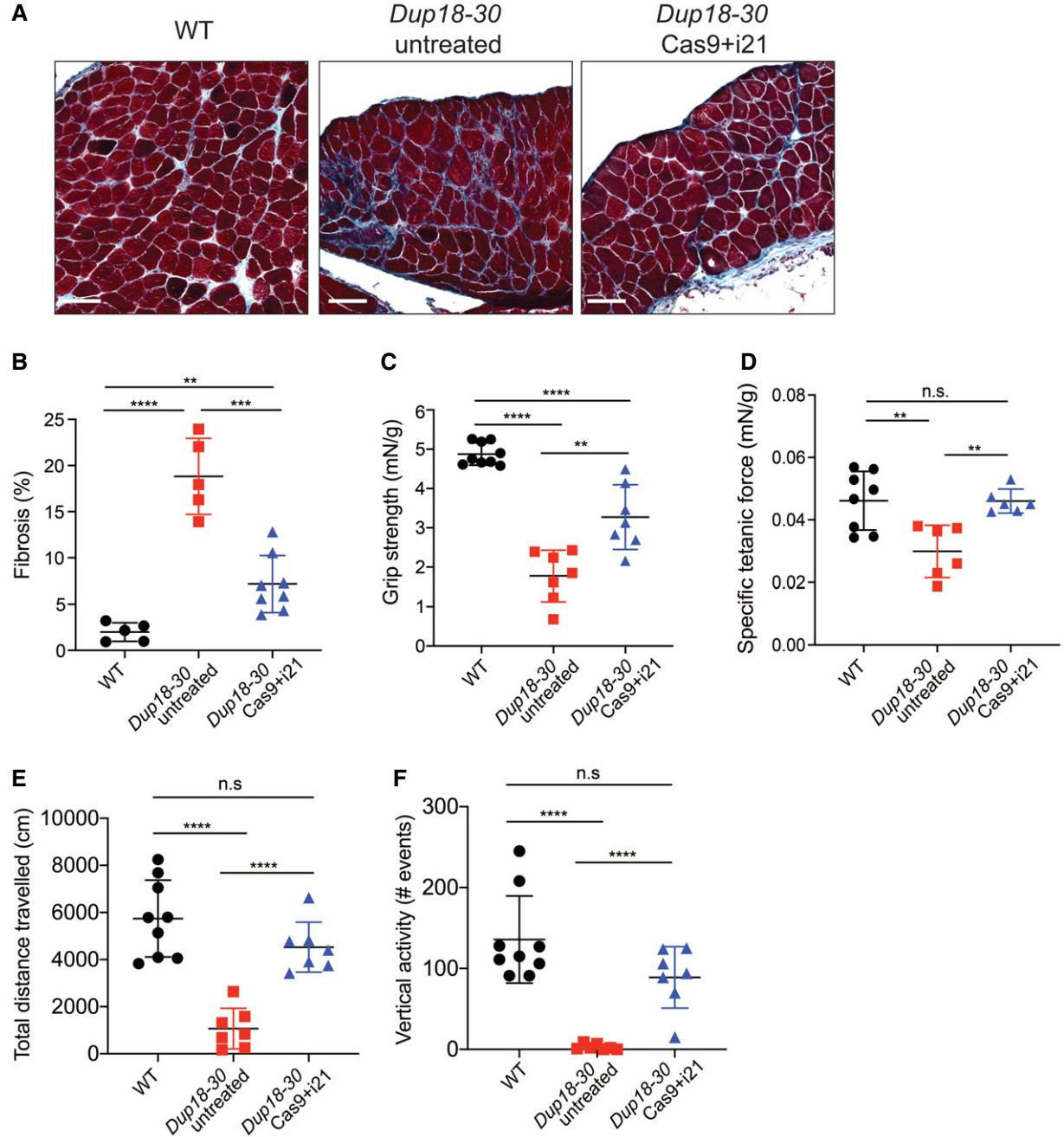

**Figure 5. The single-sgRNA/Cas9 treatment improves DMD phenotypes in *Dup18-30* mice.**

A   Fibrosis was analyzed in the diaphragm via Masson Trichrome staining. Scale bars, 100 μm.
B   Quantification of diaphragm fibrosis in WT, *Dup18-30*-untreated, and *Dup18-30*-treated mice. WT, *n* = 5; *Dup18-30* untreated, *n* = 5; *Dup18-30* treated, *n* = 8.
C–F  WT, *Dup18-30*-untreated, and *Dup18-30*-treated mice were tested 7 weeks after treatment by measuring (C) forelimbs grip strength, (D) specific tetanic force, and (E) total distance traveled and (F) vertical activity in an open-field chamber. WT, *n* = 8–9; *Dup18-30* untreated, *n* = 6–7; *Dup18-30* treated, *n* = 6–7.

Data information: All data are represented as means ± SD. Statistical analyses were performed with Student's *t*-test. n.s. not significant, **$P < 0.01$, ***$P < 0.001$, ****$P < 0.0001$.

function, and as little as 20% provides full protection of muscle fibers against exercise-induced damage (van Putten *et al*, 2012; van Putten *et al*, 2013; van Putten *et al*, 2014). In accordance with literature, the induction of full-length dystrophin expression ranging from 4 to 30% as seen in our study was sufficient to improve muscle histopathology by reducing the presence of central nuclei and decreasing the amount of fibrosis in analyzed muscles. Furthermore, the single-sgRNA approach generated phenotypic improvement,

with strength and locomotor functions being no different between treated and WT mice. Intriguingly, dystrophin-negative fibers in *Dup18-30*-treated mice showed reduced presence of central nuclei. This result supports the previously formulated hypothesis (Dunant *et al*, 2003; Zhang *et al*, 2010; Meregalli *et al*, 2015) that edited dystrophin-positive muscle fibers exert a bystander effect that benefits the overall health of the muscle tissue and significantly strengthens treatment efficacy.

Our study presents some limitations that should be considered. First, treatment outcome was evaluated with respect to only a subset of disease manifestations. In future studies, it would be important to analyze the efficacy of the treatment on skeletal muscle function by measuring eccentric muscle contraction, as dystrophic muscles are reported to be damaged preferentially during eccentric contractions (Hu *et al*, 2020). Moreover, as the majority of DMD patients develop severe cardiac dysfunction, it would be beneficial to investigate the cardiac function in treated and untreated *Dup18-30* mice. While we have shown significant restoration of full-length dystrophin expression and phenotypic improvement 7 weeks after treatment administration, it would be important to assess the long-term effects of the treatment. It has been shown that editing persists for several months in cardiomyocytes, as they do not turnover (Bengtsson *et al*, 2020). On the other hand, the cycles of necrosis and regeneration typical of dystrophic skeletal muscles might lead to loss of editing if restoration of dystrophin is not robust and if there is not sufficient targeting of muscle satellite cells. Further investigative directions should assess initiation of the treatment later in life to determine whether disease manifestations could be reversed with the current approach or if any optimization would be required. In line with what has been previously reported in other studies employing AAV-based genome-editing approaches (Amoasii *et al*, 2017; Nelson *et al*, 2019; Min *et al*, 2019a; Moretti *et al*, 2020), the single-sgRNA treatment showed no significant off-target effects. However, it would be important to repeat the off-target analysis with human-specifics sgRNAs when proceeding with the clinical development of our strategy.

Future studies following this work may apply the single-sgRNA approach to larger-scale duplications, including chromosomal multi-gene duplications. While we have previously demonstrated the success of this strategy *in vitro* in MDS patient cells, *in vivo* testing in an animal model of the disease is warranted. The optimization of our strategy may be required to enhance the efficiency of duplication removal. In this regard, some strategies may involve increasing the ratio of sgRNA to Cas9 (Min *et al*, 2019b) or considering different delivery vehicles such as nanoparticle-coated AAVs to increase *in vivo* delivery efficiency and tropism for the tissue of interest (Moretti *et al*, 2020).

In summary, we presented the first animal model faithfully recapitulating an intragenic multi-exonic tandem duplication. We subsequently demonstrated successful *in vivo* correction of a large tandem duplication using a single-sgRNA CRISPR/Cas9 approach, leading to significant phenotypic improvement with full-length dystrophin expression. Our work underscores the tremendous potential of CRISPR/Cas9 genome editing in both modeling and treating human tandem duplication mutations. These findings effectively highlight new avenues of duplication disorder research and expand the arena of emerging therapies for affected patients.

# Materials and Methods

## CRISPR sgRNAs guide design and cloning

All the sgRNAs utilized in this study were designed using https://www.benchling.com/ and selected based on minimal off-target activity. The sgRNAs designed to generate the mouse model targeted sequences flanking the regions to duplicate or invert. The sgRNAs to remove the duplication were designed targeting intronic regions included in the genomic duplication. Complementary oligo strands were annealed, phosphorylated, and cloned into the pX459 plasmid for mouse model generation, or pX601 plasmid for duplication editing. All guides utilized in this study are reported in Appendix Table S5.

## Cell culture and sgRNA screening

A panel of sgRNAs (Appendix Table S5) was designed against the introns included in the duplicated region of the *Dup18-30* mouse model. N2A cells were obtained from the American Type Culture Collection (ATCC) and cultured in DMEM (Gibco) supplemented with 10% fetal bovine serum, 1% L-glutamine, and 1% Pen-Strep (Gibco) at 37°C, 5% $CO_2$. N2A cells were seeded in a 24-well plate and transfected using Lipofectamine 2000 (Thermo Fisher Scientific). Cells were transfected with 500 ng of the px601 plasmid containing SaCas9 and the sgRNA of interest. Genomic DNA was isolated 72 h after transfection. The region encompassing each sgRNA target site was amplified using primers indicated in Appendix Table S6. The amplicons were purified using the QIAquick PCR Purification Kit (Qiagen) and Sanger sequenced using the forward primer. To test guide efficiency, the Sanger sequencing files were analyzed using the online ICE analysis tool (Synthego).

## Animal studies

All the animals utilized in this study were maintained in the specific-pathogen free facility at the Toronto Centre for Phenogenomics (TCP) on a 12-h light/dark cycle and provided with food and water *ad libitum* in individually ventilated units (Techniplast). Only male mice on a C57Bl6/J background aged up to 7 and 15 weeks of age were utilized in this study. Mice were randomly assigned to either the experimental or control group. Histopathological quantifications and behavioral analysis presented in the manuscript were performed by blind experimenters. All presented procedures involving animals were performed in compliance with the Animal for Research Act of Ontario and Guidelines of the Canadian Council on Animal Care. The TCP Animal Care Committee reviewed and approved all procedures.

## Mouse model generation

The *Dup18-30i* mouse model was generated utilizing C57BL/6NTac-C2 Embryonic Stem (ES) cell line as previously described (Gertsenstein *et al*, 2010). Briefly, $5 \times 10^6$ C2 ES cells were transfected with 4 pX459-sgRNAs plasmids (3 μg each plasmid) using Biorad Gene Pulser (250 V, 500 μF Capacity, ∞OhMs), and then plated onto 10 cm tissue culture-treated dish covered with $1 \times 10^6$ mouse embryonic fibroblasts (MEF) in FBS-DMEM (15% FBS, 2 mM

GlutaMAX™, 1 mM Na Pyruvate, 0.1 mM non-essential amino acids, 50 U/ml penicillin and streptomycin (Gibco), 0.1 mM 2-mercaptoethanol (Sigma), and 1000 U/ml LIF (EMD Millipore ESG1107)). Selection with 1 µg/ml of puromycin started at 24 h and continued for 3 days. Colonies were picked and replicated for cryopreservation and DNA analysis. Genotyping was performed via PCR followed by Sanger sequencing with the primers indicated in Appendix Table S6. The positive ES cell clones were expanded, and DNA was isolated for additional confirmation prior to aggregation. Cells were thawed in KnockOut™ Serum Replacement (KOSR)+2i media (15% KOSR, 1 mM Na Pyruvate, 0.1 mM non-essential amino acids, 0.1 mM 2-mercaptoethanol, 2 mM GlutaMAX™, 50 U/ml penicillin/streptomycin (all Gibco), 500 U/ml LIF (EMD Millipore ESG1107), 5 µg/ml Insulin (Sigma I0516), 1 µM of the mitogen-activated protein kinase inhibitor PD0325901 (StemGent 04-0006), and 3 µM of the glycogen synthase kinase-3 inhibitor CHIR99021 (StemGent 04-0004)) and passaged 2–3 times prior to aggregation.

Briefly, embryos from superovulated CD-1 (ICR) females were collected at 2.5 day post-coitum (dpc). Zonae Pellucidae of embryos were removed by the treatment with acid Tyrode's solution (Sigma). Embryos and ES cell colonies were aggregated as previously described (Gertsenstein *et al*, 2010). Aggregates were cultured overnight in microdrops of KSOM with amino acids (Zenith/Life Global) and covered by embryo-tested mineral oil at 37°C in 94% air/6% $CO_2$. The following morning, morulae and blastocysts were transferred into the uteri of 2.5 dpc pseudo-pregnant CD-1 females. Chimeras were identified at birth by the presence of black eyes and later by the coat pigmentation. Male chimeras were bred with CD-1 females and then confirmed transmitter with C57Bl6/J mice. ES cell germline transmission was confirmed by genotyping.

To correct the inversion present in the *Dup18-30i* mouse model, 2 sgRNAs were electroporated in *Dup18-30i* embryos. Briefly, homozygous *Dup18-30i* female mice were superovulated as previously described (Gertsenstein & Nutter, 2018) and were mated overnight with *Dup18-30i* hemizygous males. Oviducts were dissected to extract the fertilized embryos. Electroporation was performed as previously described (Gertsenstein & Nutter, 2018). Zygotes were briefly washed with Acid Tyrode's (Sigma T1788) to weaken the zona pellucida followed by Opti-MEM media (Thermo Fisher Scientific 31985062). The zygotes were placed into a 20 µl mixture of Cas9 RNP and sgRNAs, and the whole volume was transferred into a 1 mm-gap cuvette (BioRad, 1652089). Twelve square pulses at 30V with 1 ms pulse duration and 100 ms interval were applied using a BioRad Gene Pulser XCell electroporator. The electroporated embryos were transferred into the uteri of 2.5 dpc pseudo-pregnant CD-1 females.

### Genomic DNA isolation, PCR, and RT–PCR

Genomic DNA was isolated using the DNeasy blood and tissue kit (Qiagen) according to the manufacturer's protocol. PCR and RT–PCR were performed using DreamTaq polymerase (Thermo Fisher Scientific) or the TaKaRa LA Taq DNA polymerase (Takara) for long-range PCR according to the manufacturer's protocol. Targeted deep amplicon sequencing PCRs were performed utilizing the Q5® High-Fidelity 2X Master Mix (New England Biolabs) according

to the manufacturer's protocol. The primers utilized for PCR amplification are reported in Appendix Table S6.

### RNA isolation and quantitative PCR

Mouse tissues were sectioned in 30-µm thin slices, collected in 1.4 mm Zirconium Beads pre-filled tubes (OPS Diagnostic), and homogenized using a MagNA Lyser (Roche Diagnostic). RNA extraction was performed using Trizol reagent (Thermo Fisher Scientific) following the manufacturer's protocol. Next, 1 µg of RNA was reverse transcribed using SuperScript III reverse transcriptase (Thermo Fisher Scientific) following the manufacturer's protocol.

Quantitative PCR was performed using the fast PowerUp SYBR Green Master mix (Thermo Fisher) on a QuantStudio3 real-time PCR (Applied Biosystems). DNA and RNA editing were analyzed with the primers reported in Appendix Table S6. ΔΔCt was analyzed to assess fold changes between edited and unedited samples.

### Whole genome sequencing

DNA extracted from mouse tails was utilized for whole genome sequencing (WGS), which was performed using the Illumina HiSeq X system (San Diego, CA, USA) by The Centre for Applied Genomics (TCAG) at the Hospital for Sick Children. In brief, 400 ng of each DNA sample was used for library preparation using the Illumina TruSeq PCR-free DNA Library Prep Kit, where DNA was sonicated into an average of 350-bp fragments. A-tailed and indexed TruSeq Illumina adapters were ligated to end-repaired sheared DNA fragments before the library was amplified. Libraries were analyzed using Bioanalyzer DNA High Sensitivity chips (Agilent Technologies, Santa Clara, CA, USA) and quantified using qPCR. The libraries were loaded in equimolar quantities and pair-end sequenced on the Illumina HiSeqX platform to generate 150-bp reads. Integrative Genomics Viewer (IGV) version 2.8.2 was used for analysis with GRCm38/mm10 as the murine reference genome.

### Deep sequencing

Genomic DNA extracted from heart and TA muscles of one untreated and 3–6 Cas9 + i21-treated *Dup18-30* mice (Qiagen DNeasy Blood and Tissue kit) was amplified using oligos designed against the i21 sgRNA on-target and the top 11 off-target sites predicted by Benchling (https://www.benchling.com/). The oligos also included the Illumina flowcell binding sequences. Purified PCR amplicons were submitted for sequencing at the Donnelly Sequencing Center at the University of Toronto (http://ccbr.utoronto.ca/donnelly-sequencing-centre). Samples were quantified using Quant-iT 1X dsDNA high-sensitivity (cat # Q33232, Thermo Fisher Scientific Inc., Waltham, USA) fluorescent chemistry on a ClarioStar instrument. A second round of PCR was performed to add specific barcodes. 10 ng per sample was amplified using the Qiagen HotstarTaq Plus PCR Master Mix (cat # 203645; Thermo Fisher Scientific), as follows: 95°C for 15 min, 5 cycles of: 94°C for 30 s, 62°C for 45 s, 72°C for 1 min, and final extension at 72°C for 10 min. Final libraries were cleaned at a ratio of 0.9:1 (bead:library) using HighPrep PCR Clean-up System (MJS BioLynx Inc., cat# MGAC60250). The libraries were quantified using the Quant-iT 1X dsDNA high sensitivity (cat # Q33232, Thermo Fisher) on a

ClarioStar instrument and were diluted to 5 ng/µl. 2 µl of each dilution was run on Agilent 4200 High Sensitivity D1000 ScreenTape with High Sensitivity D1000 Reagents (cat #5067-5584 and 5067-5585, Agilent Technologies). Top stock libraries were pooled at equimolar ratios. The final pool was run on an Agilent Bioanalyzer dsDNA High Sensitivity chip (cat# 5067-4626) and quantified using NEBNext Library Quant Kit for Illumina (cat # E7630L, New England Biolabs). The quantified library pool (95%) and phiX (5%) mix were hybridized at a final concentration of 12.1 pM and sequenced with 150-bp paired-end reads on the Illumina MiSeq platform using a v2, Micro flowcell, and 300 bp read length (R1: 151, IR1: 10, IR2: 10, R2: 151). 4.3 M PF clusters had a quality score above Q30. Samples were demultiplexed according to assigned barcode sequences. FASTQ format data were analyzed using the CRISPResso2 software (Clement et al, 2019).

## Protein isolation and Western blot

Sectioned muscles collected in the Zirconium bead tubes were homogenized in 500 µl of RIPA homogenizing buffer (50 mM Tris–HCl pH 7.4, 150 nM NaCl, 1-mM EDTA, supplemented with protease-inhibitor cocktails (Roche)) and lysed with a MagNA Lyser. Subsequently, 500 µl of RIPA double-detergents buffer (2% deoxycholate, 2% NP40, 2% Triton X-100 in RIPA homogenizing buffer) was added to the lysates, which were then incubated for 45 min at 4°C with gentle agitation, then centrifuged for 10 min at 17,900 rcf. Protein concentration was measured using a BCA Assay (Thermo Scientific). Protein was separated on a 3–8% Tris-Acetate gel and transferred using an iBlot 2 transfer apparatus (Thermo Fisher Scientific). A 5% milk solution in 1× TBST was used for blocking for 1 h at room temperature. The membrane was then incubated with primary antibodies mouse monoclonal anti-dystrophin MANDYS8 (D8168, Sigma, 1:1,000), mouse anti-HA tag (ab130275, Abcam, 1:1,000), polyclonal rabbit anti-calnexin (ab22595, Abcam, 1:5,000) overnight at 4°C. This was followed by a 1-h incubation at room temperature with horseradish peroxidase-conjugated goat anti-rabbit IgG (Abcam, ab6721) or anti-mouse IgG (Abcam, ab205719). Primary and secondary antibodies were diluted in blocking solution. Signal detection was achieved using SuperSignal West Femto Maximum Sensitivity Substrate (Thermo Fisher Scientific) according to the manufacturer's protocol.

## Immunofluorescence and H&E

Muscles were sectioned at 8 µm thickness and processed for immunofluorescence analyses according to standard procedures. The muscle sections were fixed in ice cold methanol for 10 min, and then blocked for 1-h at room temperature in 3% normal goat serum (Cedarlane), 0.2% BSA (Sigma) in 1× PBS (Wisent). Primary antibodies were incubated overnight at 4°C. Primary antibodies used were rabbit polyclonal anti-dystrophin (abcam15277, Abcam, 1:200), rabbit polyclonal anti-syntrophin alpha 1 (ab11187, Abcam, 1:100), rabbit polyclonal anti-beta-sarcoglycan (ab222241, Abcam, 1:100), rabbit polyclonal anti-nNOS (1:50) previously described by Crosbie et al, 1998, and rat monoclonal anti-Laminin-2 (a2 Chain) (4H8-2, Sigma Aldrich, 1:500). Secondary antibodies used were goat polyclonal anti-rabbit Alexa Fluor 594 (ThermoFisher, 1:250) and goat polyclonal anti-rat Alexa Fluor 488 antibody (ThermoFisher, 1:250).

### The paper explained

#### Problem

Duplication mutations underly several rare inherited disorders, including Duchenne Muscular Dystrophy (DMD). Importantly, the lack of animal models genetically recapitulating duplication mutations has hampered the development of novel therapeutic options for patients affected by these disorders.

#### Results

Here, we generated the first mouse model with a multi-exonic duplication in the Dmd gene that recapitulates a patient mutation. The Dup18-30 mouse model shows typical DMD muscle histopathology and disease phenotypes. We implemented a CRISPR/Cas9-based strategy utilizing a single-sgRNA to correct the duplication mutation in vivo. Upon administration of the treatment, we showed successful removal of the duplication mutation in cardiac and skeletal muscles of the Dup18-30 mice. The treatment led to restoration of full-length dystrophin expression, as well as improvement of muscle histopathology and overall behavior in the DMD mice.

#### Impact

Our work underscores the powerful potential of CRISPR/Cas9 genome editing for both modeling and treating tandem duplication mutations. These results open new therapeutic avenues for DMD patients with duplication mutations as well as all duplication mutation disorders at large.

Primary and secondary antibodies were diluted in blocking solution. Images were acquired with a Zeiss Axiovert 200 M microscope. PerkinElmer Volocity software was used for image acquisition.

Hematoxylin and eosin (H&E) staining was conducted using a standard protocol (Kemaladewi et al, 2017).

Trichrome staining was performed at the Pathology laboratory at The Centre for Phenogenomics, Toronto (TREAT-NMD SOP MDC1A_M.1.2.003).

Slides were scanned with the 3Dhistech Panoramic 250 Flash II digital scanner and analyzed with CaseViewer software. Analysis was performed using ImageJ 1.52s software. 300 fibers obtained from 3 different field of views per muscle per animal were analyzed for central nuclei quantifications (Dubach-Powell, 2008). Fibrosis was quantified from 3 images per muscle by image segmentation.

## Virus production and injections

The px601-saCas9-sgRNA plasmid was packaged into AAV9 vectors by the Boston Children's Hospital Viral Core. For temporal vein injection into neonatal pups, a dose of $3 \times 10^{12}$ viral genomes was used. Injection volume was brought to 40 µl with 1× PBS (Gibco).

## Functional tests

Open-field activity test, grip strength, and assessment of in vivo muscle force were performed at the Toronto Centre for Phenogenomics. For the open-field test, mice were placed in the frontal center of a transparent Plexiglas open field (41.25 cm × 41.25 cm × 31.25 cm) illuminated by 200 lx. A trained operator, unaware of the nature of the projects and treatments, performed the experiments. The VersaMax Animal Activity Monitoring System recorded vertical activities and total distance traveled for 20 min per animal.

Forelimb Grip Strength was measured with the grip strength test apparatus (BIO-GS3, Bioseb). The mice were lowered over the grid keeping the torso horizontal and allowing the forepaws to attach to the grid. Then, the mice were gently pulled back by their tails and maximal grip strength was recorded. The data are shown as the average of 3 pulls, corrected to body weight.

An *in vivo* muscle-contraction test was performed using 1300A: 3-in-1 Whole Animal System and analyzed using dynamic muscle control/analysis (DMC/DMA) high-throughput software suite (Aurora Scientific). The mice were anesthetized by an intraperitoneal injection of a ketamine/xylazine cocktail at 100 mg/kg and 10 mg/kg of body weight, respectively. Percutaneous electrodes were placed in the tibialis anterior, and specific tetanic force at 175 Hz was recorded and corrected according to body weight.

### Statistical analysis

All statistical analyses were performed using GraphPad Prism (GraphPad software). Data normality was tested using Shapiro–Wilk test. Two-tailed Student's *t*-tests were performed to evaluate significant differences between two groups. The exact numbers of animals used in each analysis are presented as individual values in dot-plot graphs, and data are presented as average ± S.D. $P < 0.05$ was considered to be significant. The exact *P*-values are reported in Appendix Table S7. For the experiments performed in this study, mice were randomly assigned to either control or experimental group. All functional and behavioral analysis and histopathological quantifications were performed by blind experimenters.

## Data availability

Data are available from the Dryad depository at https://doi.org/10.5061/dryad.66t1g1k1d and at the NCBI BioProject database (https://www.ncbi.nlm.nih.gov/bioproject) under accession number PRJNA688584 (https://www.ncbi.nlm.nih.gov/bioproject/?term=pRJNA688584).

Expanded View for this article is available online.

## Acknowledgments
We thank Zahra Baghestani, Antonia Knoth, Vanessa Gomes, and Gina Desatnik for their technical support in this study. The Cohn laboratory members are gratefully acknowledged for their input in this study. We thank Marina Gertsenstein and Monica Pereira (Model Production Core, TCP) for their support with mouse model generation. We also thank Igor Vukobradovic and Ann Fleniken (Clinical Phenotyping Core, TCP), and Chris Rand (Aurora Scientific) for their assistance with mouse behavioral testing. We thank Milan Ganguly (Pathology Core, TCP) for his assistance with histological stains. We thank Sergio Pereria and Bhooma Thiruv (The Centre for Applied Genomics) for their support with whole genome sequencing analysis. We thank Tanja Durbic (Donnelly Sequencing Centre) for her support with deep sequencing analysis. Anti-nNOS AP rbt 200 was a kind gift from Dr. Kevin Campbell (HHMI, University of Iowa). The visual abstract of the paper was created with Biorender.com. This work was funded by the McArthur family Foundation (R.D.C.), Jesse's Journey (R.D.C.), Duchenne UK (R.D.C.), Duchenne Research Fund (R.D.C.), and the Michael Hyatt Foundation (R.D.C.). E.M. was funded by the Ermenegildo Zegna Founder's Scholarship, and D.W. was funded by the Restracomp Award (SickKids).

## Author contributions
Conceptualization: EM, DW, EAI, and RDC. Methodology: EM, DW, SLE, AF, TWYW, KL, EH, SZR, SV, AC, MS, and MR. Formal analysis: EM, DW, SLE, OS, and SZR. Investigation: EM, DW, SLE, AF, TWYW, KL, EH, SZR, SV, AC, MS, and MR. Provide resources: EAI and RDC. Data curation: EM, DW, EAI, and RDC. Writing, original draft preparation: EM. Writing, reviewing, and editing: EM, DW, OS, SLE, KL, SZR, EAI, and RDC. Supervision and Project administration: EAI and RDC. Funding acquisition: EM, DW, EAI, and RDC. All authors reviewed the final version of the manuscript.

## Conflict of interest
The authors declare that they have no conflict of interest.

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
