## [Review Process File · EMBO Molecular Medicine]

Targeted genome editing in vivo corrects a Dmd duplication restoring wild-type dystrophin expression

Eleonora Maino, Daria Wojtal, Sonia Evagelou, Aiman Farheen, Tatianna Wai Ying Wong, Kyle Lindsay, Ori Scott, Samar Rizvi, Elzbieta Hyatt, Matthew Rok, Shagana Visuvanathan, Amanda Chiodo, Michelle Schneeweiss, Evgueni Ivakine, and Ronald Cohn

DOI: [10.15252/emmm.202013228](https://doi.org/10.15252/emmm.202013228)

Corresponding author: Ronald Cohn (ronald.cohn@sickkids.ca)

Review Timeline:

Submission Date:	4th Aug 20
Editorial Decision:	17th Sep 20
Revision Received:	31st Dec 20
Editorial Decision:	27th Jan 21
Revision Received:	6th Feb 21
Accepted:	10th Feb 21

Editor: Zeljko Durdevic

Transaction Report:

17th Sep 2020

Dear Dr. Cohn,

Thank you for the submission of your manuscript to EMBO Molecular Medicine, and please accept my apologies for the delay in getting back to you. We have received feedback from two of the three reviewers who agreed to evaluate your manuscript. Given that referee #2 will unfortunately not be able to return his/her report in a timely manner, and that both referees #1 and #3 are overall positive, we prefer to make a decision now in order to avoid further delay in the process. Should referee #2 provide a report, we will send it to you, with the understanding that we will not ask for an additional revision. As you will see from the reports below, both referees are positive and find the study interesting and important. However, they also raise some concerns that should be addressed in a major revision of the current manuscript.

Addressing the reviewers' concerns in full will be necessary for further considering the manuscript in our journal. Acceptance of the manuscript will entail a second round of review. Please note that EMBO Molecular Medicine encourages a single round of revision only and therefore, acceptance or rejection of the manuscript will depend on the completeness of your responses included in the next, final version of the manuscript. For this reason, and to save you from any frustrations in the end, I would strongly advise against returning an incomplete revision.

We realize that the current situation is exceptional on the account of the COVID-19/SARS-CoV-2 pandemic. Therefore, please let us know if you need more than three months to revise the manuscript.

I look forward to receiving your revised manuscript.

Yours sincerely,

Zeljko Durdevic

***** Reviewer's comments *****

Referee #1 (Remarks for Author):

This study addresses the absence of animal models for intragenic duplications, an important objective to allow testing of genome editing therapies for potential use in a range of human disorders, including DMD. They successfully generated an intragenic tandem duplication model for

DMD in mice and provide clear step-by-step descriptions of how this novel model was obtained. Further, genome editing of the duplication restores some expression of the wildtype protein, leading to functional improvements in the mouse model. This is an interesting and novel research project that will be of interest to the wider scientific community.

Major comments

1. The authors spent a lot of time and effort to generate the DMD duplication mouse model and it appears that an unwanted recombination event led to an additional step in this procedure. While this is an important lesson and the exact technical details are relevant to the expert reader, the full details are a bit of a detraction to the main message of the study. Could most of this be moved to the supplement?
2. The authors show morphological improvement (reduced central nuclei) and increased dystrophin expression in muscle. It would be of interest to see whether on a single-fibre level the reduction in central nuclei coincides with dystrophin expression. Previous studies (both gene transfer and transgenic models) have shown a bystander effect, that even fibers, which remain dystrophin negative, benefit from dystrophin expression in adjacent fibers. Could the authors provide this type of analysis?
3. The most sensitive functional assay for dystrophin deficiency is force measurements with eccentric contraction. This assay has not been carried out by the authors as far as we can see. The authors may want to highlight this point on the limitation section in the discussion.
4. In many cases, the histology images are rather small and suboptimal quality. Can the authors provide better images?
5. Did the authors carry out any functional analysis of the diaphragm, which is the muscle in mdx mice that is affected earlier and more severe than limb muscles?

Minor comments:

1. An explanation of what intragenic tandem duplications are would be useful in the introduction
2. It would be helpful for Fig 3B to have labels pointing out features mentioned in the text - eg presence of fibrosis and central nuclei
3. Define SaCas9 at first use
4. Figure 4F and 4G - what does 'WT 25% and 50%' refer to? Loading of 25%/50% protein concentration as compared to Dup18-30 mice?
5. In the results section it is stated that treated mice show rescued expression of DGC components, including alpha-syntrophin and beta-sarcoglycan - were other proteins tested? If so it would be good to include this data in the supplementary figure (S5A) along with the extra muscles tested. Further, a comment is made about the levels of restoration correlating with the level of dystrophin - was this analysis performed? Do you see the same % increase in dystrophin per mouse/per tissue as you do for the other DGC components? Or is there just a general trend of increase? Either include the results of this analysis or reword the sentence.
6. As tetanic force measurements are included in the main manuscript for characterisation of the mouse model I would propose including Figure S5D in to the main manuscript too.
7. In the results text it is stated that 'contractile force was 43% higher, compared to untreated mice ($p < 0.05$) (Fig S5D)' - however, in the graph shown in Fig S5D the change is labelled as 'not significant'. Please make this clear in the text.
8. Were any other off-target structural changes identified (e.g. in other genes) using WGS after treatment of mice with the Cas9/gRNAs?

9. In the methods section please add in the solution used for diluting primary and secondary antibodies for western blots.

10. In the methods section does '300 fibres analysed from 3 different fields of view per animal' refer to 3 fields of view per muscle per animal?

11. For the in vivo muscle contraction test it is stated that plantar flexor muscles are stimulated but the results text refers only to the TA muscle: 'The muscle strength of the Dup18-30 mice was evaluated with grip strength and specific tetanic force measurements in the TA muscle' and 'To correlate the above findings, we evaluated grip strength and contractile force measurements in the TA.' - which muscles were used for this study? Furthermore, it would be helpful to reword these sections to make it clear that the grip strength test was to measure forelimb strength and not the TA muscle.

12. In the statistical analysis section there is no reference to whether the distribution of data was determined prior to selecting the students t-test - were all data sets normally distributed?

Referee #3 (Comments on Novelty/Model System for Author):

This is an interesting article describing the generation of a murine dystrophin duplication disease model. There are several existing models (nonsense, exon deletion and one single exon duplication) that allow testing of various therapies (cell, gene replacement, exon skipping etc) but this is the first multi-exon duplication model.

Hence, while directly relevant to this type of mutation in the dystrophin gene, the technology could be applied to other large genomic duplications .. ie some small whole genes

The generation of the model highlighted the challenges with this technology including the unexpected consequences (ie generating the duplication while inducing a multi-exon inversion immediately downstream (which was subsequently corrected using this technology).

Novelty must be regarded as high, medical impact is rated as medium as this is relevant for a relatively rare type of dystrophin mutation (many different duplications spread across the gene) but potential applications to other regions/genes raises the impact.

The adequacy of the model I have classified as N/A as this is essentially a unique mutation that may or may not reflect the human case (it certainly will not at the genetic level with different breakpoints and then the additional re-inversion of exons 31-34). Nevertheless, this is still a relevant mouse dystrophinopathy model.

Referee #3 (Remarks for Author):

The manuscript by Maino and colleagues describes the construction of a dystrophin multi-exon duplication in the mouse and then the use of single guide RNA CRISPR/Cas9 correction strategy.

While there are other mouse dystrophinopathy models, I believe this is the first multiple-exon duplication model. It was somewhat ironic that in generating the exon 18-30 duplication an inversion was induced that needed to be corrected.

I have a few comments and questions that the authors could address to further strengthen this manuscript.

The authors mention that a Dup2 mouse model does not recapitulate the nature of tandem duplications at the genomic level. Perhaps this could be reworked. The dup2 mouse is a tandem duplication but I am not sure if there is "one nature" for intragenic duplications. As with dystrophin deletions, the break-points are different between different (unrelated) families with the same deletion at the RNA level. I am certain the same would be true for duplications.

As for animal models, the mouse is convenient to use but does not recapitulate the more serious or obvious nature of the human condition.

The authors mention complete absence of dystrophin and refer to Fig 2C (a western). Was there any evidence of revertant fibres? These are rare dystrophin positive fibres observed in the original mdx mouse (nonsense mutation in exon 23) and arose from large natural multi-exon skipping events spanning exons 15-30(ish)

The authors only briefly compare their Dup18-30 mice to the WT mice and mention comparing to the exon 23 nonsense mice. Could that be expanded to compare the dystrophinopathy models?

Could the selection of the gRNAs be expanded, why was intron 21 chosen, how large is this intron compared to the others.

The most impressive feature of the application of this technology to multi-exon duplications is the potential to generate normal dystrophin.

The immunostaining clearly shows dystrophin expression but is the staining of treated section in Figures 4 and 5 somewhat patchy. It would have been good to include the WT sections for comparison (as done for Figure3).

I am not sure if it is feasible (and not necessary for this paper) but would it be possible to use the single sgRNA in cultured cells, clonally expand and confirm the changes that are occurring at the DNA level during the removal of the duplication.

Referee #1 (Remarks for the Authors)

This study addresses the absence of animal models for intragenic duplications, an important objective to allow testing of genome editing therapies for potential use in a range of human disorders, including DMD. They successfully generated an intragenic tandem duplication model for DMD in mice and provide clear step-by-step descriptions of how this novel model was obtained. Further, genome editing of the duplication restores some expression of the wildtype protein, leading to functional improvements in the mouse model. This is an interesting and novel research project that will be of interest to the wider scientific community.

We would like to thank the reviewer for the encouraging comments regarding the novelty of the presented results and overall positive comments on the manuscript. In the current resubmission we have addressed most of the comments made by the reviewer. Detailed point-by-point responses are listed below.

Major comments

1. The authors spent a lot of time and effort to generate the DMD duplication mouse model and it appears that an unwanted recombination event led to an additional step in this procedure. While this is an important lesson and the exact technical details are relevant to the expert reader, the full details are a bit of a detraction to the main message of the study. Could most of this be moved to the supplement?

We now reorganized the mouse model generation figures in the paper. We have only one figure in the main manuscript (Fig 1) which summarizes the 2-step generation strategy we adopted to generate the *Dup18-30* mouse model and its molecular characterization. The detailed characterization of the abnormal splicing and dystrophic phenotype of the *Dup18-30i* mouse model have been moved to the supplementary Appendix Fig S2-S3.

2. The authors show morphological improvement (reduced central nuclei) and increased dystrophin expression in muscle. It would be of interest to see whether on a single-fibre level the reduction in central nuclei coincides with dystrophin expression. Previous studies (both gene transfer and transgenic models) have shown a bystander effect, that even fibers, which remain dystrophin negative, benefit from dystrophin expression in adjacent fibers. Could the authors provide this type of analysis?

We have now included this analysis in Fig EV4. In the *Dup18-30* untreated mice, about 57.8% of the dystrophin-negative fibers have central nuclei. In the treated mice, only 14.7% of the dystrophin-negative fibers present one or more central nuclei, indicating a reduction in 75.4% of central nuclei in dystrophin-negative fibers post-treatment. These results support the concept of previously published hypotheses that gene therapy treatments can exert a bystander effect in which overall dystrophin restoration improves muscle health.

3. The most sensitive functional assay for dystrophin deficiency is force measurements with eccentric contraction. This assay has not been carried out by the authors as far as we can see. The authors may want to highlight this point on the limitation section in the discussion.

We included this point in the limitations paragraph in the discussion section of the paper.

4. In many cases, the histology images are rather small and suboptimal quality. Can the authors provide better images?

We improved the resolution of all histology images in the paper and reacquired all images present in the editing section of the paper.

5. Did the authors carry out any functional analysis of the diaphragm, which is the muscle in mdx mice that is affected earlier and more severe than limb muscles?

Though we did not carry out any functional assays for the diaphragm muscle, we agree on the significance of showing treatment efficacy in this muscle. We have now strengthened the manuscript by including histological analysis for the diaphragm muscle. We included histological images and central nuclei quantification in Fig 4B-C. This analysis highlighted a 68.5% reduction of central nuclei in the *Dup18-30* Cas9 treated mice compared to untreated mice. Moreover, we performed Masson Trichrome staining to quantify the degree of fibrosis (Fig 5A-B). Fibrosis was reduced by 61.8% in the treated mice compared to controls, decreasing from 18.8% to 7.2% in the treated mice. Taken together, these results show a beneficial effect of the single-sgRNA treatment on the diaphragm.

Minor comments:

1. An explanation of what intragenic tandem duplications are would be useful in the introduction. The definition of intragenic duplications has now been included in the introduction.

2. It would be helpful for Fig 3B to have labels pointing out features mentioned in the text - eg presence of fibrosis and central nuclei

We included labels to point out fibrosis and central nuclei in Fig 2B (previously Fig 3B).

3. Define SaCas9 at first use

We defined SaCas9 when first introduced.

4. Figure 4F and 4G - what does 'WT 25% and 50%' refer to? Loading of 25%/50% protein concentration as compared to Dup18-30 mice?

Yes, we loaded 25% and 50% of WT protein to use as references in the Western Blots. This has now been clarified in Fig 3 F-G (previously Fig 4 F-G) and in the Fig EV3A-B legend.

5. In the results section it is stated that treated mice show rescued expression of DGC components, including alpha-syntrophin and beta-sarcoglycan - were other proteins tested? If so it would be good to include this data in the supplementary figure (S5A) along with the extra muscles tested. Further, a comment is made about the levels of restoration correlating with the level of dystrophin - was this analysis performed? Do you see the same % increase in dystrophin per mouse/per tissue as you do for the other DGC components? Or is there just a general trend of increase? Either include the results of this analysis or reword the sentence.

We now included staining for neuronal nitric oxide (nNOS) in Tibialis Anterior (TA), triceps and diaphragm of WT, *Dup18-30* treated and *Dup18-30* untreated mice in Fig EV3B.

We have not performed any analysis to correlate the levels of dystrophin expression with the levels of restoration of DGC components. Overall, we see a generalized trend of restoration of DGC components in all treated mice. As such, we removed the confusing sentence from the paper results.

6. As tetanic force measurements are included in the main manuscript for characterisation of the mouse model I would propose including Figure S5D in to the main manuscript too.
Tetanic force measurement has been included in the main manuscript (Fig 5D).

7. In the results text it is stated that 'contractile force was 43% higher, compared to untreated mice ($p < 0.05$) (Fig S5D)' - however, in the graph shown in Fig S5D the change is labelled as 'not significant'. Please make this clear in the text.

We would like to thank the reviewer for pointing this oversight out. We have included an additional cohort of mice for all of the functional analyses carried out in the paper, including contractile force measurements. In our final data, tetanic force was 48% higher in *Dup18-30* Cas9 treated compared to *Dup18-30* untreated mice ($p < 0.01$). No significant statistical difference was observed between *Dup18-30* treated mice and WT.

8. Were any other off-target structural changes identified (e.g. in other genes) using WGS after treatment of mice with the Cas9/gRNAs?

We did not perform WGS in the *Dup18-30* mice after treatment. We want to point out, however, that the relatively low coverage of WGS (about 30X) would very unlikely detect any rare events caused by non-specific editing considering that the on-target editing is less than 10% (Appendix Figure S4C-D)

We do, however, agree on the significance of testing for off-target changes post-treatment, and to address this point, have performed Next-gen targeted DNA sequencing of amplicons for the top 11 predicted off-target sites. The extremely high coverage of this technique (where every amplicon is sequenced on average 60000X) would allow us to effectively identify any possible rare non-specific editing event. Notably, we did not detect any non-specific editing at any off-target location from our analyses, which are now included in Appendix Table S4.

9. In the methods section please add in the solution used for diluting primary and secondary antibodies for western blots.

We updated the methods section with the solutions utilized for the dilutions.

10. In the methods section does '300 fibres analysed from 3 different fields of view per animal' refer to 3 fields of view per muscle per animal?

We clarified this in the methods section.

11. For the in vivo muscle contraction test it is stated that plantar flexor muscles are stimulated but the results text refers only to the TA muscle: 'The muscle strength of the *Dup18-30* mice was evaluated with grip strength and specific tetanic force measurements in the TA muscle' and 'To

correlate the above findings, we evaluated grip strength and contractile force measurements in the TA.' - which muscles were used for this study? Furthermore, it would be helpful to reword these sections to make it clear that the grip strength test was to measure forelimb strength and not the TA muscle.

As the reviewer correctly pointed out we measured tetanic force in the TA and grip strength only in the forelimb muscles. We have now clarified the results and methods sections referring to these measurements.

12. In the statistical analysis section there is no reference to whether the distribution of data was determined prior to selecting the students t-test - were all data sets normally distributed?

We analyzed the normality of the datasets prior to selecting t-test utilizing the Shapiro-Wilk test (most powerful normality test). All the data presented in this paper are normally distributed. We have now added this information in the statistics paragraph in the methods section of the paper.

Reviewer 3

Referee #3 (Comments on Novelty/Model System for Author):

This is an interesting article describing the generation of a murine dystrophin duplication disease model. There are several existing models (nonsense, exon deletion and one single exon duplication) that allow testing of various therapies (cell, gene replacement, exon skipping etc) but this is the first multi-exon duplication model.

Hence, while directly relevant to this type of mutation in the dystrophin gene, the technology could be applied to other large genomic duplications .. ie some small whole genes

The generation of the model highlighted the challenges with this technology including the unexpected consequences (ie generating the duplication while inducing a multi-exon inversion immediately downstream (which was subsequently corrected using this technology).

Novelty must be regarded as high, medical impact is rated as medium as this is relevant for a relatively rare type of dystrophin mutation (many different duplications spread across the gene) but potential applications to other regions/genes raises the impact.

The adequacy of the model I have classified as N/A as this is essentially a unique mutation that may or may not reflect the human case (it certainly will not at the genetic level with different breakpoints and then the additional re-inversion of exons 31-34). Nevertheless, this is still a relevant mouse dystrophinopathy model.

Referee #3 (Remarks for Author):

We would like to thank the reviewer for appreciating the novelty and impact of the presented work.

Detailed point-by-point responses to the reviewer comments are listed below.

he manuscript by Maino and colleagues describes the construction of a dystrophin multi-exon duplication in the mouse and then the use of single guide RNA CRISPR/Cas9 correction strategy.

While there are other mouse dystrophinopathy models, I believe this is the first multiple-exon duplication model. It was somewhat ironic that in generating the exon 18-30 duplication an inversion was induced that needed to be corrected.

I have a few comments and questions that the authors could address to further strengthen this manuscript.

The authors mention that a Dup2 mouse model does not recapitulate the nature of tandem duplications at the genomic level. Perhaps this could be reworked. The dup2 mouse is a tandem duplication but I am not sure if there is "one nature" for intragenic duplications. As with dystrophin deletions, the break-points are different between different (unrelated) families with the same deletion at the RNA level. I am certain the same would be true for duplications.

The *Dup2* has been an excellent and helpful mouse model, and can be even used in studying certain therapeutic approaches, such as exon skipping. However, although the *Dup2* mouse model recapitulates the exon 2 duplication on the *Dmd* transcript level, on the genomic level it resembles more of a "retro-transposition"-like event rather than a tandem duplication event (see Fig.1A, Vulin et al., 2015). Indeed, in the *Dup2* model a small 0.5 kb region containing murine *Dmd* exon 2 and surrounding intronic regions were inserted an ~32 kb upstream of the endogenous exon 2. In tandem duplications large (>10 kb) highly homologous (or even identical) sequences are adjacent to each other in the same orientation. This unique feature of tandem duplications might influence a DNA repair outcome after single sgRNA-mediated duplication removal strategy. Due to this limitation, our strategy could not have been adequately tested using the existing *Dup2* mouse. Nevertheless, we agree that both models are valid and useful in different ways, and have now clarified this point in the introduction of the paper.

As for animal models, the mouse is convenient to use but does not recapitulate the more serious or obvious nature of the human condition.

The authors mention complete absence of dystrophin and refer to Fig 2C (a western). Was there any evidence of revertant fibres? These are rare dystrophin positive fibres observed in the original mdx mouse (nonsense mutation in exon 23) and arose from large natural multi-exon skipping events spanning exons 15-30(ish)

We quantified the presence of revertant fibers in the TA and triceps of *Dup18-30* mice with IF at 15 weeks of age (Fig 2C). Only 3-5% fibers showed dystrophin expression. We hypothesize that the levels of dystrophin expression by revertant fibers are too low to be detected by Western Blotting.

The authors only briefly compare their Dup18-30 mice to the WT mice and mention comparing to the exon 23 nonsense mice. Could that be expanded to compare the dystrophinopathy models?

We have now included in the first paragraph of the discussion a more expanded description of the phenotype of the *Dup18-30* mouse model and a comparison with the available dystrophin-deficient mouse models.

Could the selection of the gRNAs be expanded, why was intron 21 chosen, how large is this intron compared to the others.

We included the explanation and results (Appendix Table S3) of the guide selection process in the results section of the paper. To select the single-sgRNA to treat the *Dup18-30* mice, we scanned all of the introns included in the duplicated region. We selected the best guide RNAs based on their off-target score and verified that they were not interfering with any predicted splicing site. The top eight-ranking guides were tested *in vitro* in N2A cells, and the most active guide was utilized for the *in vivo* treatment. The size of intron 21 has not been taken into consideration for guide design.

The most impressive feature of the application of this technology to multi-exon duplications is the potential to generate normal dystrophin.

The immunostaining clearly shows dystrophin expression but is the staining of treated section in Figures 4 and 5 somewhat patchy. It would have been good to include the WT sections for comparison (as done for Figure3).

As suggested, we have included WT control staining for all of the histological and immunohistochemical images presented in the manuscript.

I am not sure if it is feasible (and not necessary for this paper) but would it be possible to use the single sgRNA in cultured cells, clonally expand and confirm the changes that are occurring at the DNA level during the removal of the duplication.

We find the point raised by the reviewer of great significance and interest. We currently have an ongoing project in the laboratory aimed at generating *in vitro* cell models of tandem duplication mutations and investigating the in-depth mechanism of duplication removal by the single-sgRNA approach.

To begin answering this question *in vivo*, we have performed target deep amplicon sequencing at the i21 gRNA on-target site and included this analysis in Appendix Figure S4 of the manuscript. We identified the presence of small indels at the predicted guide cut site, leading to about 3-5% editing in the TA and heart of *Dup18-30* treated mice.

However, this assay presents some limitations: 1) our analysis likely underestimates the editing percentage, as perfect joining after duplication removal would be considered a non-editing event; 2) the assay does not discriminate between events happening upon removal of the duplication and cutting events that do not lead to duplication removal.

In addition, we wanted to investigate whether the removal of the duplication could cause inversions similar to what was reported for the mouse model generation. We designed a PCR-

based assay to detect the presence of inversions of the excised fragment, but did not identify any detectable inversion event.

27th Jan 2021

Dear Dr. Cohn,

Thank you for the submission of your revised manuscript to EMBO Molecular Medicine. I am pleased to inform you that we will be able to accept your manuscript pending the following final amendments:

1) With the beginning of the new year, we encountered high number of submissions, so that our data editors were not able to process all received manuscripts. Therefore, we will send you the document with data editor's suggestions as soon as our data editors process your manuscript. Please do not submit your revised manuscript before we send you the file with data editor's suggestions. Thank you for your understanding.

2) In the main manuscript file, please do the following:

- Add up to 5 keywords.
- Make sure that all special characters display well.
- Remove the font color.
- Add callouts for Fig 1E and 2H. Remove the callout for Fig EV1E as this panel does not exist. Please check and make sure that all figure references in the main text correspond to the respective figure.
- In M&M, statistical paragraph should reflect all information that you have filled in the Authors Checklist, especially regarding randomization, blinding, replication.
- In M&M, for animal work, confirm that all experiments were performed in accordance with relevant guidelines and regulations. The manuscript must include a statement in the Materials and Methods identifying the institutional and/or licensing committee approving the experiments and the licensing number when appropriate. Gender, age, origin of the animals and genetic background must be indicated, along with housing conditions.
- Indicate in legends exact $p=$ values, not a range, along with the statistical test used. To keep the figures "clear" some authors found providing an Appendix table Sx with all exact p -values preferable. You are welcome to do this if you want to.

3) Appendix: Please check and make sure that all appendix tables are labeled "Appendix Table S1" etc. in the legends and in the manuscript text.

4) The Paper Explained: Please provide "The Paper Explained" and add it to the main manuscript text. Please check "Author Guidelines" for more information.

<https://www.embopress.org/page/journal/17574684/authorguide#researcharticleguide>

5) For more information: There is space at the end of each article to list relevant web links for further consultation by our readers. Could you identify some relevant ones and provide such information as well? Some examples are patient associations, relevant databases, OMIM/proteins/genes links, author's websites, etc...

6) As part of the EMBO Publications transparent editorial process initiative (see our Editorial at <http://embomolmed.embopress.org/content/2/9/329>), EMBO Molecular Medicine will publish online a Review Process File (RPF) to accompany accepted manuscripts. This file will be published in conjunction with your paper and will include the anonymous referee reports, your point-by-point response and all pertinent correspondence relating to the manuscript. Let us know whether you agree with the publication of the RPF and as here, if you want to remove or not any figures from it prior to publication. Please note that the Authors checklist will be published at the end of the RPF.

7) Please provide a point-by-point letter INCLUDING my comments as well as the reviewer's reports and your detailed responses (as Word file).

I look forward to reading a new revised version of your manuscript as soon as possible.

Yours sincerely,

Zeljko Durdevic

***** Reviewer's comments *****

Referee #1 (Remarks for Author):

The authors have responded to all my comments. They have provided further experimental data that strengthen their study.

Referee #3 (Comments on Novelty/Model System for Author):

first generation and description of a multi-exon duplication in the mouse dystrophin gene, and successful gene editing correction.

A very nice piece of work and I have only down-graded novelty and impact to medium as this is a rare type of mutation. However, the technology could presumably be relevant to other gens and conditions

Referee #3 (Remarks for Author):

The authors have addressed all my comments/questions

I am happy for this to be published in its current form

The authors performed the requested editorial changes.

We are pleased to inform you that your manuscript is accepted for publication and is now being sent to our publisher to be included in the next available issue of EMBO Molecular Medicine.

Corresponding Author Name: Ronald Cohn

Manuscript Number: EMM-2020-13228